# Bridging the Intention–Behavior Gap in Organic Food Consumption: Empirical Evidence from China

**DOI:** 10.3390/foods13142239

**Published:** 2024-07-16

**Authors:** Yaqin Liu, Peng Wang, Mengya Zhang, Xi Chen, Ke Li, Jianying Qu

**Affiliations:** 1Key Laboratory of Computing and Stochastic Mathematics (Ministry of Education), School of Mathematics and Statistics, Hunan Normal University, Changsha 410081, China; lyq@hunnu.edu.cn (Y.L.); 13488516512@163.com (P.W.); zmy1224@hunnu.edu.cn (M.Z.); chenxi1024@hunnu.edu.cn (X.C.); likekent1208@163.com (K.L.); 2Hunan Institute for Carbon Peaking and Carbon Neutrality, Hunan Normal University, Changsha 410081, China; 3School of Mathematics and Statistics, Hunan First Normal University, Changsha 410205, China

**Keywords:** organic food consumption, theory of planned behavior, structural equation model, intention‒behavior gap, multinomial logistic regression

## Abstract

The transition to a low-carbon economy emphasizes the importance of green and low-carbon consumption; yet, there is often a discrepancy between consumers’ intentions and their actual behavior regarding organic foods. This study aims to identify strategies to bridge this gap. The research model of organic food consumption intention and behavior is constructed, and a structural equation model is used to test the research hypotheses based on a valid sample of 480 residents of Guangdong Province through an online questionnaire survey. Further, the intention‒behavior gap is defined and its determinants are investigated through multiclass logistic regression. Finally, we categorize and forecast the alignment between consumption intentions and behaviors using machine learning algorithms. The results reveal that attitudes, social interactions, and cognitive information play crucial roles in aligning intentions with behaviors. By enhancing social information exchange or improving cognitive understanding, consumers can reduce their intention‒behavior discrepancy. This research offers valuable policy recommendations for fostering green consumption among residents from various perspectives.

## 1. Introduction

As environmental damage and resource depletion continue to escalate, green consumption has emerged as a promising sustainable solution. The Annual Report on Action to Address Climate Change, issued by the Chinese Academy of Social Sciences, highlights that China’s per capita greenhouse gas emissions of carbon dioxide equivalent stand at 8.2 tons, with approximately 4.3 tons attributed to consumption, representing approximately 52%. Among household consumption-related carbon emissions, food contributes 1.1 tons, ranking second. Presently, there is a global surge in green consumption due to a heightened awareness of environmental issues, particularly due to the increased consumption of organic food. Organic food is not only healthier but also promotes environmentally friendly consumption.

Green low-carbon consumption is the key means to reduce carbon emissions at the end of the economy. However, there is a huge gap between green consumption intention and behavior. In the world of green consumerism, organic products play a central role. For research on the relationship between organic food consumption intention and behavior, scholars generally believe that intention has a positive impact on behavior. Using the theory of planned behavior, alphabet theory, and other behavior theories for reference, Yadav confirmed that the intention to consume organic food had a positive impact on related consumption behavior [1]. In addition, some studies have described the mediating role of intention in behavior formation. Existing studies have divided the participants into four quadrants as follows: low intention and low behavior, low intention and high behavior, high intention and low behavior, and high intention and high behavior. Because intention has a significant positive impact on behavior, consumers with low intention and low behavior can promote consumer behavior by improving their intention. It is worth noting that there is also inconsistency between intention and behavior, which is mainly manifested such that consumers with high intention and low behavior are common, while consumers with low intention and high behavior are very rare. Scholars at home and abroad have actively explored the topics of organic food and green consumption. Most studies are rooted in the theory of planned behavior, and often ignore important factors such as social interaction and cognitive information. In addition, the literature using structural equation modeling has identified some factors that jointly affect consumer intentions and behaviors [2]. Some researchers use consistency to measure the gap between intention and behavior, and use a regression model or machine learning technology to predict the consistency of consumption intention and behavior. However, these studies have not accurately determined the mechanism of the transition from intention to behavior.

In light of this, this study focuses on the factors that influence intention and behavior in organic food consumption, aiming to explore strategies to bridge the gap between intention and behavior. Specifically, the study first constructs a conceptual framework of organic food consumption intention and behavior, and the model is verified by an online survey conducted among a random sample of 480 Guangdong residents using structural equation modeling. Subsequently, the study employs a multiple logistic regression model to analyze the degree of influence of different factors on the intention‒behavior gap. Finally, machine learning techniques are utilized to predict the gap between intention and behavior in organic food consumption, and the key contributing factors to these predictions are analyzed. We find that the main influencing factors in adjusting intention and behavior are attitude, social interaction, and cognitive information, and there is difference between the influencing factors in different consumer groups.

The potential contributions of this study include (1) integrating social interaction and cognitive information into the theoretical framework of planned behavior and examining their impact on organic food consumption intention and behavior and (2) quantitatively investigating the intention‒behavior gap, thereby revealing the mechanism through which intention translates into behavior in the context of organic food consumption.

The structure of this paper is organized as follows: Section 2 provides a summary of the theoretical foundations and a relevant literature review. Section 3 includes the construction of the theoretical model and the development of the measurement scale. Section 4 presents an empirical analysis of the model, examining the influencing factors of organic food consumption intention and behavior. Section 5 concludes the study with future research directions.

## 2. Literature Review and Research Hypotheses

### 2.1. Literature Review

Although the research literature on green low-carbon consumption has been very rich, the intention–behavior gap of organic food consumption is paid less attention. This part first analyzes the relationship between intention and behavior, then analyzes the influencing factors of intention and behavior, and finally comments on the existing research on the gap between intention and behavior.

#### 2.1.1. Relationship between Intention and Behavior

At present, the research on the relationship between intention and behavior has been widely used in green consumption, organic food consumption, and other fields. In theory, intention is considered to be an immediate antecedent of behavior [3]. Based on rational behavior theory, alphabet theory, cognitive behavior theory, etc., researchers have confirmed that organic food consumption intention has a positive impact on consumption behavior [1,4,5]. The concept of green consumption has been widely recognized, and a large number of scholars have used questionnaires to measure consumers’ intentions and behavior. The analysis results find that the mean value of consumers’ intentions is always greater than the mean value of behavior, and there is a huge inconsistency between consumers’ intentions and behavior [6,7,8].

For the inconsistency between intention and behavior, researchers generally use the theory of planned behavior to explain it, but the theory ignores the interference factors between intention and behavior, that is, the restrictions of various conditions in real life. Groening et al. proposed that future research on green consumption should focus on social interaction factors when studying green marketing [9]. In addition, Rana et al. believe that the inconsistency between intention and behavior needs to consider environmental cognition, food quality and safety, and other factors [10].

#### 2.1.2. Influence Path of Intention on Behavior

There are many influencing factors of consumption intention and consumption behavior, and the related research can be divided into psychological factors and non-psychological factors. From the perspective of psychological factors, consumers’ attitudes, perceptions, risk awareness, health awareness, and environmental concerns have a positive effect on behavior [1,11,12,13,14], but insufficient cognition will hinder consumer behavior [8]. Cognitive behavior is also affected by the personal characteristics of consumers. Generally speaking, the cognitive level of female consumers and young consumers is more likely to affect their behavior [15]. Consumers with high education are important groups for the purchasing of organic food [16,17,18], and have more consumption knowledge and stronger green consumption awareness [19].

In addition, the information dissemination between people, the frequency of social interaction, and the scale of social networks all have a significant impact on green consumption behavior. The social pressure that individuals feel for a specific behavior will have an impact on individual behavior [20,21]. Therefore, improving descriptive subjective norms is conducive to maintaining a high level of consumption behavior [22]. Among many influencing factors, the development of incentive mechanisms or the increase in environmental protection publicity and other positive information dissemination can significantly improve the low-carbon consumption intention of consumers [23]. In addition, product attributes and functions will directly and indirectly affect behavior [24]. High prices or insufficient consumer awareness are the main reasons that hinder consumers’ green consumption. Consumers have a significant preference for organic food with additional attributes such as green food certification, organic food certification, and origin information [25].

#### 2.1.3. Research on the Gap between Intention and Behavior

Research methods used to test the gap between consumption intention and behavior include structural equation models [2], multiple regression models [26], and machine learning [27,28]. Among them, a structural equation model is the mainstream research method. Most research conclusions show that enhancing the sense of responsibility can effectively narrow the gap between intention and behavior, while perceived behavior control and environmental participation can bridge the intention‒behavior gap. Furthermore, multiple regression models are often used to analyze the reasons for the gap between intention and behavior. Using a binary regression model, researchers divided this gap into consistent and inconsistent. Their study found that income, education level, environmental concerns, and carbon emission knowledge had a significant impact on the gap between low-carbon intention and behavior [29]; the main obstacles to the inconsistency between intention and behavior are the high price, low availability, or inconvenience of the purchase of organic products in the market [30]. In addition, machine learning algorithms have been used to classify and predict intention, behavior, and their gap [31,32,33].

There are also a few scholars who use econometric methods for analysis, such as using a random parameter logit regression method to study consumers’ preferences for different attributes of safety-certified products and factors affecting safety consumption. One study found that green food certification, organic food certification, origin information, and other attributes have a significant impact on the gap [13].

In general, the current research on organic food and green consumption has achieved some meaningful results, but there are still some deficiencies. (1) From the perspective of the influencing factors of intention and behavior, most studies introduce many factors to extend the theory of planned behavior, and make this theory apply to the field of organic food consumption, but less consideration is given to social interaction and cognitive information. (2) From the perspective of the research status of the gap between intention and behavior, most studies have found some factors that jointly affect consumers’ intentions and behavior using structural equation models. A few researchers describe the intention–behavior gap as consistent, utilize binary regression model to conduct consistent classification, and use machine learning to predict whether it is consistent. However, these studies cannot accurately identify the transformation from intention to behavior.

In view of this, this paper adds social interaction factors and social cognitive factors to the theory of planned behavior to analyze the impact of social interaction and social cognitive factors on consumption intention and behavior. At the same time, based on the quantitative analysis of the gap between intention and behavior, a machine learning algorithm is used to realize the multiclassification regression and multiclassification prediction of the consistency of intention and behavior.

### 2.2. Research Hypotheses

The main theories considered in this study include the theory of planned behavior [1], social interaction theory [34], and social cognitive theory [30]. According to the theory of planned behavior, intention directly influences behavior, while a number of factors largely affect behavior indirectly through intention. Social interaction theory posits that people can change their perceptions through social interactions, and individual interdependence is influenced by other individuals. According to this theory, decision makers’ preferences, expectations, and budget constraints are affected by the behavior of others. Social cognitive theory suggests that individual behavior, cognition, and other individual characteristics interact with the external environment to determine human activities.

Based on the theory of planned behavior, a structural equation model is constructed to examine the formation mechanism of consumers’ intentions and behavior. Subjective norms, attitudes, and perceived behavior control are identified as the key positive factors influencing consumers’ intentions and behavior. From a sociological perspective, social interaction theory has shown that social interaction significantly impacts low-carbon consumption behavior [34]. Additionally, based on social cognition theory, Pieniak concluded that a lack of cognitive information hinders consumers’ consumption behavior [35]. Finally, considering the attributes of organic food itself, according to the research, the attributes of organic food have a positive moderating effect on intention and behavior, and this study selects two main attributes to study the moderating effect of price fairness and purchase convenience. A summary of the classification of hypotheses is shown in Table 1, which identifies their characteristics and reference sources.

**H1a** . *Subjective norms positively influence consumption intention*.

**H1b.** *Subjective norms positively affect consumption behavior*.

**H2a.** *Attitude positively influences consumption intention*.

**H2b.** *Attitude positively affects consumption behavior*.

**H3a.** *Perceived behavioral control positively influences consumption intention*.

**H3b.** *Perceived behavior control positively affects consumption behavior*.

**H4a.** *Social interaction positively influences consumption intention*.

**H4b.** *Social interaction positively affects consumption behavior*.

**H5a.** *Cognitive information positively affects consumption intention*.

**H5b.** *Cognitive information positively affects consumption behavior*.

**H6.** *Consumption intention positively affects consumption behavior*.

**H7.** *The influence of consumption intention on consumption behavior is positively moderated by price fairness*.

**H8.** *The influence of consumption intention on consumption behavior is positively moderated by the purchase convenience*.

Considering the assumptions mentioned above, this paper constructs a model to examine the influencing factors of organic food consumption intention and behavior, as depicted in Figure 1. From the perspective of individual consumers, a structural equation model is employed to analyze the formation mechanism of organic food consumption intention and behavior, as well as the strength of the relationships between variables. Furthermore, the impact of social interaction and cognitive information on consumer behavior is considered. Additionally, the attributes of organic food itself are taken into account. Previous research has shown that the attributes of organic food have a positive regulatory effect on the intention to engage in behavior. Accordingly, this study focuses on two main attributes, namely, price fairness and purchase convenience, to investigate their regulatory effects.

## 3. Research Methods

In this paper, a questionnaire survey is conducted among residents in Guangdong Province, China, who are over the age of 18. This study aims to analyze the influencing factors of consumption intention and consumption behavior and explore the reasons for the gap between the two. The scales used in the questionnaire are well established.

First, a presurvey is administered to test the reliability and validity of the questionnaire. Then, a formal questionnaire is developed and distributed through the Questionnaire Star platform. Finally, effective questionnaires are selected for analysis.

After obtaining the effective questionnaires, a structural equation model is constructed using Amos. The hypothesis proposed in this paper is verified by grouping the demographic variables. Subsequently, sample data suitable for analyzing the gap between intention and behavior are screened, and the transformation from intention to behavior is explored using multiclass logistic regression.

### 3.1. Questionnaire Design

The questionnaire consists of two parts: (1) basic information, including gender, age, education level, monthly family income, and city, and (2) variables, including attitude, perceived behavior control, subjective norms, social interaction, cognitive information, price fairness, purchase convenience, intention, and behavior. Except for demographic variables, all items in the questionnaire are measured using a five-point Likert scale. The item design for the variables is presented in Table 2.

### 3.2. Data Collection and Processing

The respondents in this survey were consumers over the age of 18 in Guangdong Province, China. After a preliminary survey, a formal questionnaire survey was conducted in late July 2022. The questionnaires were distributed via the Questionnaire Star platform, and a total of 582 questionnaires were collected. After eliminating 102 invalid questionnaires, 480 valid questionnaires were obtained, resulting in an effective response rate of 82.5%. The distribution of samples based on demographic characteristics is relatively balanced and suitable for grouping research. The basic characteristics of the sample are outlined in Table 3.

## 4. Empirical Results

### 4.1. Examination of Scales

#### 4.1.1. Reliability and Validity

Before conducting statistical analysis, it is essential to test the reliability and validity of the formal scale. The results of the reliability and validity tests are presented in Table 4. The findings indicate that the Cronbach’s alpha values for all variables are above 0.7, indicating the high reliability and validity of the questionnaire scale. The KMO value and Bartlett spherical test results showed a significant statistical value (*p* = 0.000), demonstrating certain structural validity and correlation among factors, which allows for further factor analysis.

Exploratory factor analysis (EFA) revealed five common factors, accounting for 71.3% of the total explanatory variance, which is greater than the threshold of 60%. The factor loading of the items is above the suggested threshold value 0.3, indicating the good discriminant validity and convergent validity of the scale.

Confirmatory factor analysis (CFA) was conducted to examine the construct validity of the model, as indicated in Table 4. Convergence validity was evaluated using average variance extraction (AVE), where the AVE values were found to be greater than 0.5, suggesting high convergence validity. Composite reliability (CR) was used to verify the reliability of survey data. The CA values in this study are all above 0.7, which represents the reliability.

#### 4.1.2. Correlation

The correlation between variables, presented in Table 4, demonstrated good discriminant validity, with the square root of the AVE surpassing the correlation between variables. The median value of the scale is three, which is taken as the cut-off point. Based on the survey results, the average intention to consume organic food is 3.941 > 3, indicating a generally high intention among consumers. On the other hand, the average consumption behavior of organic food is 2.927 < 3, suggesting relatively low consumption behavior among consumers. This reveals a discrepancy between intention and behavior. Moreover, the standard deviation of intention is less than the standard deviation of behavior, implying that there are minor differences in intention and substantial differences in behavior among different consumers.

Table 4 implies that the correlations among the variables are all significant. Among the five independent variables, the strongest correlation is observed between subjective norms and attitude, followed by social interaction and attitude. This highlights the importance of attitude as an independent variable. Comparing the correlation of the independent variables with the dependent variables, it can be concluded that social interaction exhibited a strong correlation with intention and behavior, signifying its significant impact on consumers’ purchase of organic food.

Furthermore, the formal scale can only be analyzed by a structural equation model if it passes the normality test. Otherwise, data processing or the use of a least squares structural equation model is necessary. The results of the normality distribution test indicated that the absolute value of the kurtosis coefficient of each item was also less than 10, and the absolute value of the skewness coefficient of each item less than three. Therefore, the data can be considered to approximately follow a normal distribution, meeting the requirements for a structural equation model.

This study integrates social interaction and cognitive information into the theory of planned behavior. Through reliability and validity analysis, seven variables are retained: subjective norms, attitudes, perceived behavior control, social interaction, cognitive information, price fairness, and purchase convenience. The correlation coefficients between social interaction, cognitive information, and the other three variables are approximately 0.351–0.501, which are less than the square root of AVE, indicating good discriminant validity. Therefore, social interaction and cognitive information (*p* < 0.01) can be incorporated into the theoretical model of organic food consumption intention and behavior.

### 4.2. Hypothesis Testing

Before testing the hypotheses, the suitability of the model is assessed. The results, shown in Table 5, indicate a good fit for each index. The chi-square ratio per degree of freedom (CMIN/DF) value falls within the range of 1–3. The goodness-of-fit index (GFI), comparative fit index (CFI), and incremental fit index (IFI) values are all above 0.9, with a threshold of 0.8 considered acceptable. The root mean square error of residual (RMR) value is less than 0.05. The root mean square error of approximation (RMSEA) and adjusted goodness-of-fit index (AGFI) values are both within the threshold requirements. The results indicate that the model is properly set for structural equation model analysis.

The results of the hypothesis test are presented in Table 6. Each path in the table is significant, confirming the acceptance of hypotheses H1–H6 proposed earlier. The most influential factor on consumption intention is social interaction (β = 0.264, *p* < 0.001), followed by attitude (β = 0.257, *p* < 0.001) and cognitive information (β = 0.230, *p* < 0.001). This suggests that consumption intention for organic food is influenced not only by consumers’ attitudes but also by social interaction and cognitive information. Directly changing consumers’ attitudes is more challenging, but modifying information interactions within society is relatively easy. The most influential factor on consumer behavior is social interaction (β = 0.173, *p* < 0.001), followed by attitudes (β = 0.154, *p* < 0.001) and subjective norms (β = 0.152, *p* < 0.001). Social interaction has a strong impact on intention and behavior, suggesting that it can effectively narrow the gap between intention and behavior.

### 4.3. Multi-Group Analysis

This study divides participants into different groups based on gender (male and female), age (young: 18–40 years old; middle-aged and elderly: 41 years old and above), education level (low: junior college and below; high: undergraduate and above), monthly household income (low and high), and city classification (first-tier and non-first-tier). The classification of cities as first-tier or non-first-tier cities is based on the 2022 ranking list of urban commercial charm by First Finance and Economics. The model fitness is evaluated, and hypothesis testing is conducted. The fitness results are presented in Table 7. It can be seen that there is a good fitness of the multi-group structural model.

Multi-group analyses by gender, age, and education level provided valuable findings, shown in Table 8. According to the significance degree of the *p*-value, the influence of each factor can be judged. In regard to male consumers, their intentions and behavior are primarily influenced by social interaction (**β** = 0.333, *p* < 0.001) and cognitive information (**β** = 0.244, *p* < 0.001). In contrast, for female consumers, intention is influenced primarily by subjective norms (**β** = 0.266, *p* < 0.01) and attitudes (**β** = 0.256, *p* < 0.01), whereas behavior is influenced mainly by social interaction and attitudes. These results indicate that male consumers are more susceptible to the influence of information exchanged between individuals, while female consumers are more influenced by their own attitudes.

The consumption intentions of young consumers are influenced primarily by cognitive information (**β** = 0.308, *p* < 0.001), while behavior is influenced by subjective norms. On the other hand, middle-aged and elderly consumers’ consumption intentions and behaviors are influenced mainly by social interaction (**β** = 0.322, *p* < 0.001). This suggests that young consumers tend to have more information available to them and are more vulnerable to external information and their own cognition, while older consumers have less interaction with information and are more susceptible to the influence of information exchanged between individuals.

For consumers with lower education levels, their consumption intentions and behavior are primarily influenced by social interaction (**β** = 0.327, *p* < 0.001). In contrast, consumers with higher education levels are influenced primarily by their own consumption attitudes (**β** = 0.268, *p* < 0.001) and cognitive information (**β** = 0.267, *p* < 0.001) in regard to consumption intention. This implies that consumers with lower education levels are more susceptible to external information, while consumers with higher education levels are comparatively more rational and influenced by their extensive knowledge.

The findings, as shown in Table 8 (continued), indicate that for consumers with low family income, their consumption intention is primarily influenced by social interaction (**β** = 0.321, *p* < 0.001) and their own attitudes (**β** = 0.313, *p* < 0.001). For consumers with high family income, their consumption intention is influenced mainly by their own consumption attitudes (**β** = 0.248, *p* < 0.01) and cognitive information (**β** = 0.239, *p* < 0.001), and their behavior is influenced mainly by social interaction (**β** = 0.212, *p* < 0.001) and perceived behavior control (**β** = 0.149, *p* < 0.01). These results suggest that consumers with lower income have weaker self-control abilities and are more vulnerable to external information, while consumers with higher income have stronger control abilities but are also affected by information exchange.

For consumers residing in first-tier cities, their consumption intentions and behaviors are influenced mainly by social interaction (**β** = 0.343, *p* < 0.001; **β** = 0.156, *p* < 0.01) and attitudes (**β** = 0.280, *p* < 0.001; **β** = 0.167, *p* < 0.01). On the other hand, consumers in non-first-tier cities are influenced primarily by social interaction (**β** = 0.293, *p* < 0.001; **β** = 0.234, *p* < 0.001) and subjective norms (**β** = 0.215, *p* < 0.01; **β** = 0.227, *p* < 0.001) in regard to consumption intention and behavior. These findings indicate that regardless of the consumer’s location, they are influenced by information exchange. However, consumers in first-tier cities have more freedom to make consumption choices based on their abilities, while consumers in non-first-tier cities are more influenced by the people around them, such as friends and family.

### 4.4. Moderating Effect Test

If the relationship between two variables is influenced by a third variable, the third variable is considered a mediating variable and has a moderating effect. The moderating effect can be tested in two ways: first, by testing the significance of the change in the F value when the model is altered; second, by testing the significance of the interaction terms in the model. This study utilizes hierarchical adjustment regression to analyze the moderating effects of price fairness and purchase convenience on consumption intention and separately analyzes the effects on purchase frequency and purchase proportion. The results of the analysis are presented in Table 9 and Table 10.

The results indicate that the influence of consumption intention on purchase frequency and purchase proportion is significantly moderated by price fairness. The interaction between consumption intention and price fairness is also significant, with a positive coefficient (β = 0.119, *p* < 0.01; β = 0.224, *p* < 0.01), suggesting that price fairness positively moderates the influence of consumption intention on purchase frequency and purchase proportion. The moderation effect of purchase convenience on the relationship between consumption intention and purchase frequency is not significant (*p* > 0.01), but it is significant when analyzing purchase proportion (β = 0.229, *p* <0.01). This suggests that purchase convenience positively moderates the influence of consumption intention on purchase proportion. Therefore, it can be concluded that price fairness positively regulates both consumption intention and consumption behavior, supporting hypothesis H7. The positive moderating effect of purchase convenience on the relationship between consumption intention and consumption behavior partially supports hypothesis H8.

### 4.5. Multinomial Logistic Regression

To analyze the gap between intention and behavior, this study categorizes the gap into three levels of consistency: inconsistency, low consistency, and high consistency. Inconsistency refers to the largest gap between intention and behavior. To narrow this gap, inconsistency needs to be transformed into low consistency, and low consistency needs to be transformed into high consistency. Previous studies have divided intention and behavior into four quadrants to determine the transformation process. However, this study conducts a more detailed analysis of the gap between intention and behavior through quantitative settings; a visualization of this process is shown in Figure 2.

The specific steps are described as follows:(1)Calculation of intention and behavior values. Considering that the intention value may be different according to the different items, this study selected the intention value with the smallest value of the three items (willing to buy, plan to buy, and buy in the future). In other words, intention values are equal to the minimum of the three items. The frequency of consumption behavior is used to represent the behavior values. Intention and behavior are qualitative variables and they range from 1 to 5.(2)Screening samples. Considering that most people’s intention values are larger than behavioral values, data are excluded to reduce the bias when intention values are less than behavior values or intention values are less than three.(3)Re-marking. Intention is categorized as low, medium, or high. Data with a value less than 3 are recorded as low, data with a value of 3–4 are recorded as medium, and data with values of 4–5 are marked as high. Behaviors are similarly categorized into low behavior, medium behavior, and high behavior. To facilitate the follow-up work, the difference between intention and behavior values can be measured by numerical distance ranging from zero to two, where zero represents inconsistency, one represents low consistency, and two represents high consistency.(4)Consistency classification. The gaps are divided into inconsistency, low consistency, and high consistency. Data with medium and high intention and low behavior are recorded as inconsistent, data with medium and high intention and medium behavior are recorded as low consistency, and data with high intention and high behavior are recorded as high consistency.

After these four steps, a total of 410 samples are selected and divided into inconsistent (61), low consistency (170), and high consistency (179) groups. Multiple logistic regression is conducted to analyze the relationships among these three categories, and each variable shows a significant positive relationship. The regression results are shown in Table 11.

Compared to inconsistency, the main factors affecting low consistency are attitude, subjective norms, and perceived behavior control, all of which are psychological variables of consumers. This suggests that changing consumer psychology can lead to a shift from inconsistency to low consistency. The main factors affecting high consistency are attitude, social interaction, and cognitive information. The latter two factors are influenced by society, highlighting the importance of information communication among people in achieving high consistency. Price fairness and purchase convenience do not play a significant role, so these two variables are excluded from the grouping regression analysis.

After grouping individuals by gender, age, and educational level, multiple logistic regression is conducted to analyze the impact of these variables. The results are presented in Table 12. The table reveals that attitude consistently emerges as the most influential variable; the data corresponding to one line of attitudes are significant. However, due to the difficulty of changing attitudes, this analysis primarily focuses on the impact of other variables.

For gender groups, on the one hand, male consumers can enhance their level of control to achieve consistency (β = 1.205, *p* < 0.01; β = 1.913, *p* < 0.01). When the gap between intention and behavior is small, they can further narrow the gap through social interaction (β = 1.967, *p* < 0.01). On the other hand, female consumers should increase their social interaction (β = 1.118, *p* < 0.01; β = 2.351, *p* < 0.01) and cognitive information (β = 0.831, *p* < 0.05; β = 1.734, *p* < 0.01), and minimize the influence of others on their normative behaviors to narrow this gap.

Regarding the age groups, younger consumers tend to engage more in social interaction (β = 1.493, *p* < 0.01; β = 2.886, *p* < 0.01) and cognitive information (β = 1.052, *p* < 0.01; β = 1.877, *p* < 0.01). However, psychological factors such as subjective norms (β = 1.555, *p* < 0.01 β = 2.26, *p* < 0.01) and perceived behavior control (β = 1.630, *p* < 0.01; β = 2.392, *p* < 0.01) have the most significant influence on narrowing this gap. Elderly consumers can narrow this gap by strengthening social interaction (β = 2.100, *p* < 0.01) and cognitive information (β = 1.197, *p* < 0.05).

Within the educated group, less educated consumers cannot effectively improve inconsistency by intensifying social interaction (β = 0.741, *p* < 0.01; β = 3.243, *p* < 0.01). However, strengthening social interaction can contribute to the transformation from low to high consistency. Furthermore, the gap can be altered by improving their own level of control (β = 0.823, *p* < 0.05; β = 1.836; *p* < 0.01). Highly educated individuals typically exhibit a high level of control (β = 2.489, *p* < 0.05; β = 3.220, *p* < 0.01). Therefore, narrowing the gap primarily involves enhancing information interaction (β = 3.443, *p* < 0.05; β = 4.201, *p* < 0.01) and cognitive level (β = 3.433, *p* < 0.01; β = 3.129, *p* < 0.01).

The results obtained from grouping by family income are shown in Table 12 (continued). Although low-income families may have poor self-control, self-control does play a role in promoting the transition from low consistency to high consistency. Narrowing the gap between intention and behavior can be achieved through information interaction and cognitive information dissemination. In contrast, high-income families primarily narrow the gap through psychological factors such as subjective norms and perceived behavior control.

### 4.6. Multiclassification Prediction Based on Machine Learning

Different supervised machine learning algorithms are utilized in this study to achieve multiclassification prediction of intention and behavior consistency. The objective is to identify the best prediction method. The variables that displayed significant consistency through multiple regression are selected as the dimensions for constructing a multiclassification prediction model. The models included decision tree, random forest, AdaBoost, and logistic regression algorithms. Specific parameters are set for each model to enable multiclassification prediction. The evaluation indicators for each model are presented in Table 13.

The prediction accuracy of the model is greater than 74% at an acceptable level of fit. This demonstrates the feasibility of the gap definition in this study, as well as the ability of the model to accurately distinguish between the three categories of consistency. Among the four models, random forest performed the best as a prediction model, and it shows that random forest has better generalization and better classification results when processing experimental datasets, followed by Logistic regression prediction model. The random forest method was used to highlight the relative significance of different features, and the feature importance analysis is illustrated in Figure 3. The three most important characteristics influencing the distinction between intention and behavior consistency are social interaction, attitude, and perceived behavior control. These variables enable accurate differentiation. Attitude and social interaction are also key factors affecting the level of consistency, as indicated by the regression results.

## 5. Conclusions and Recommendations

Through the above analysis, the following three conclusions are drawn:(1)Enhancing the social interaction and cognitive information levels of consumers is the most effective means of improving their intention and behavior and narrowing the intention–behavior gap. By leveraging psychological variables such as subjective norms and perceived behavior control, or social variables such as social interaction and cognitive information, it is possible to effect changes in intention and behavior. Social interaction has a stronger impact, followed by cognitive information, both of which play vital roles in narrowing the gap.(2)The influencing factors of the intention and behavior gap vary among different consumer groups. Male consumers, young consumers, and less educated consumers are susceptible to external information, and the gap can be reduced by strengthening social interaction. The disparity between the desires and behaviors of female consumers and elderly consumers is relatively small. Female consumers are more influenced by subjective norms, while elderly consumers are primarily affected by information interaction. Highly educated consumers tend to be more rational and knowledgeable, making it easier to narrow the gap through cognitive information. Most high-income residents in first-tier cities have greater freedom and possess strong perceptual control ability, allowing them to rely on their own means to consume. By reinforcing social interaction and enhancing cognitive information, the gap can be diminished. Conversely, consumers in non-first-tier cities generally have lower income levels and are more influenced by individual psychological factors. They need to adjust their own norms and enhance their perceptual control ability to bridge this gap.(3)Machine learning algorithms have the capability to execute consistent multiclassification predictions. The prediction accuracy can reach 80%, which confirms the feasibility of classification. The best prediction algorithm to utilize is random forest. Both attitude and social interaction are key features in the prediction process. By utilizing these two features, the level of consistency among consumers can be effectively distinguished. Consequently, the level of the intention and behavior gap can be roughly assessed based on consumers’ attitudes and social interaction.

Based on the aforementioned conclusions, the following three policy suggestions are proposed from the perspectives of marketers, consumers, and the government.

From the perspective of marketers, it is essential to formulate differentiated marketing strategies tailored to each consumer group. For young male consumers with lower education levels, it is crucial to implement diverse promotional activities to increase their organic food consumption rate. We can actively hold organic food-related events, and the public can participate in reasonable pricing, and they can taste organic food. For middle-aged and elderly consumers with higher education levels, they can take the initiative to introduce information and the benefits of organic food. Guidance should be provided to encourage purchase intentions during the shopping process. Consumers residing in non-first-tier cities with lower income levels require the establishment of a strong sense of identity toward organic food consumption. This can be achieved through consistent messaging and establishing a clear standard such as promoting the environmental value and nutritional value of organic food.

From the perspective of consumers, it is recommended to separate organic food from other food products to induce a change in consumption habits, and put the location of organic food in the place where consumers can most easily reach it. Consumers should also have access to expanded sales channels, reasonable pricing, the setting of different price zones, and the implementation of more scientific, refined, and efficient management practices. Furthermore, organizing organic food in distinct areas with different lighting and smells can help create awareness and emphasize its accessibility to consumers.

From a government perspective, it is crucial to establish effective information dissemination policies to enhance consumer awareness of organic food. Organic food should be promoted through market management and environmental protection publicity, the construction of circulation infrastructure should be strengthened, and a well-known brand should be created. Counterfeit organic products on the market are severely punished by law, so that consumers can buy with confidence. The standardized production system and institutional certification system should be strengthened to strengthen consumer confidence. Government departments should cooperate with relevant institutions to popularize science to the public, so that more knowledge of organic food can be circulated among the public, and consumers’ awareness of health and environmental protection can be aroused. Government departments can collaborate with relevant research institutions to increase public knowledge regarding the types and benefits of organic food. This will help raise public awareness about the importance of health and environmental protection.

This paper presents an empirical analysis of questionnaires to identify the main factors influencing consumer intention, behavior, and gaps. However, it is important to note that the survey focused only on the consumption attitudes and behaviors of residents over 18 years old in Guangdong Province; although the amount of data grouped is sufficient, other significant results may be obtained by increasing the amount of data. Thus, the results may not be applicable at the national level. In order to better understand the specific situation of the gap between consumer intention and behavior in China, it is suggested that the research area should be expanded, the heterogeneity of different regions should be analyzed, and precise policies should be formulated. In addition, due to the limitation of sample characteristics, only the consistency of intention and behavior is divided into three levels: inconsistency, low agreement, and high agreement; in order to avoid the impact on the research, the sample data with an intention value smaller than the behavior value are excluded. In order to obtain more detailed research results, the gap can be more refined, that is, the level of division can be increased, but the requirements for the accuracy of the data and the amount of data will also be more stringent, so the questionnaire should also be changed to a seven-level scale or a higher scale to obtain more accurate sample data. In addition, increasing the sample size will also help to obtain more reliable findings. At the same time, it is necessary to conduct expert interviews to verify the conclusions of the study to ensure that the conclusions are accurate and valid.

## Figures and Tables

**Figure 1 foods-13-02239-f001:**
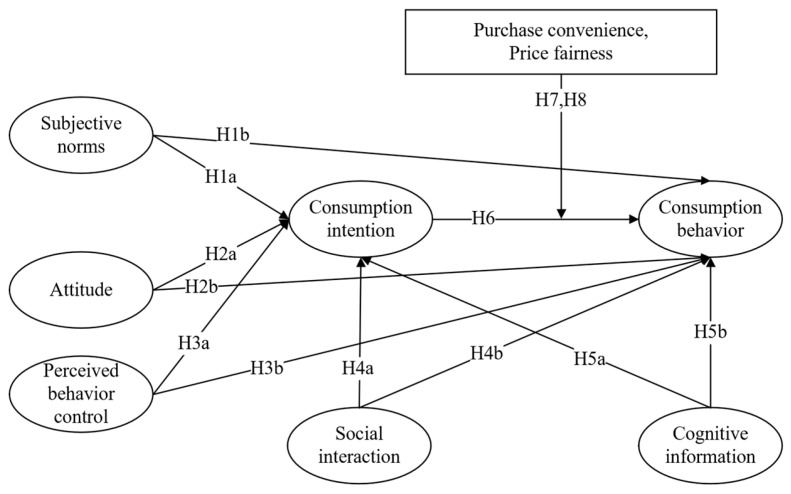
A model of the influencing factors on organic food consumption intention and behavior.

**Figure 2 foods-13-02239-f002:**
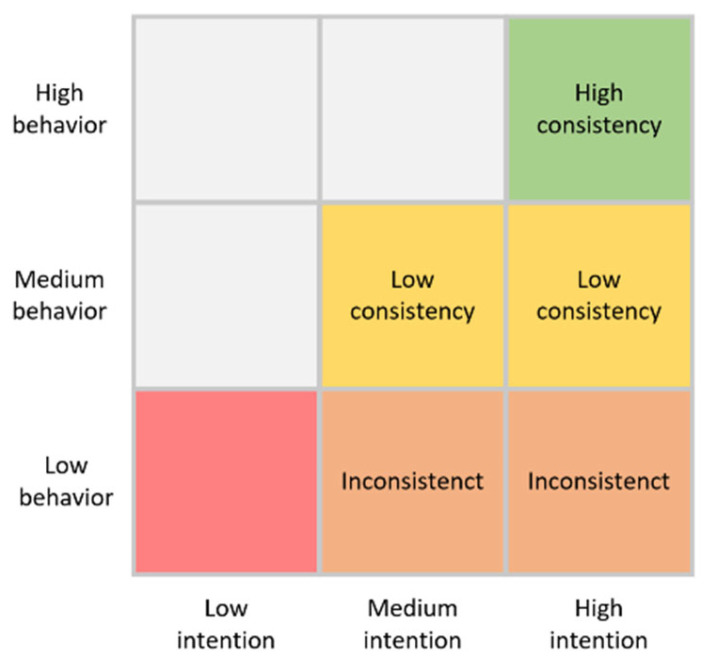
Consistency classification between intention and behavior.

**Figure 3 foods-13-02239-f003:**
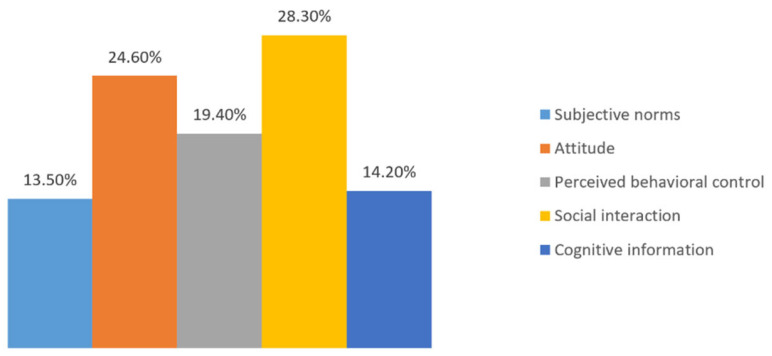
Feature importance analysis for random forest.

**Table 1 foods-13-02239-t001:** Classification and description of assumptions.

Category	Hypothesis	Reference	Instructions
Hypothesis of planned behavior theory	H1–H3, H6	[1,3,36,37]	The assumed relationship between the original variables involved in the theory of planned behavior
Extended hypothesis	H4, H5	[21,34,36,37,38,39]	A hypothetical relationship between variables added to the theory of planned behavior
Moderating effect hypothesis	H7, H8	[13,30,37,40]	A hypothetical relationship that acts directly on intention to affect action

**Table 2 foods-13-02239-t002:** Description of variables and measurement items.

Variables	Measurement Items	Sources
Subjective norms (SNs)	Government policies and media campaigns will lead me to buy organic food	[1]
	My friends and significant others buy organic food	
Attitude (ATT)	Buying organic food is a good idea	[3]
	Buying organic food is a smart choice	
	Buying organic is the right thing to do	
	Buying organic food can be enjoyable	
Perceived behavior control (PBC)	Buying organic food is easy for me	[1]
	There are many channels to buy organic food and it is convenient to buy	
	There is a lot of publicity and advertising for organic food	
Social interaction (SI)	How many of your neighbors keep in touch with you	[21]
	How many relatives keep in touch with you	
	How many friends keep in touch with you	
	You discuss the frequency of food-related consumption with your relatives	
	You and your friends discuss the frequency of food-related consumption	
	You discuss the frequency of food-related consumption with your neighbors	
Cognitive information (CI)	I’m a health-conscious consumer	[36,38]
	I’m concerned about food quality and safety	
	I’m concerned about environmental pollution	
	I am concerned about the environmental problems caused by the use of pesticides	
Price fairness (PF)	The price of organic food is reasonable	[13]
	The price of organic food is fair	
	The price of organic food is acceptable	
Purchase convenience (PC)	In my neighborhood, there are supermarkets that sell organic food	[40]
	The supermarkets and chain stores I frequent have an ample selection of organic foods	
	Organic food is easily available in supermarkets and chain stores	
Consumption intention (IN)	I prefer to buy organic food	[3]
	I plan to buy organic food	
	I will buy organic food in the future	
Consumption behavior (BE)	How often you currently buy organic food	[36]
	The percentage of your current purchases of organic food	

**Table 3 foods-13-02239-t003:** Demographic characteristics of the sample.

Demographic Variables	Categories	Frequency	Percentage
Gender	Male	234	48.8%
	Female	246	51.2%
Age (years)	18–25	62	12.9%
	26–40	210	43.8%
	41–50	156	32.5%
	>50	52	10.8%
Educational level	High school or below	68	14.2%
	Junior college	83	17.3%
	Undergraduate	303	63.1%
	Graduate	26	5.4%
Monthly household income (CNY)	Less than 5000	47	9.8%
	5000–10,000	123	25.6%
	10,001–20,000	167	34.8%
	20,001–40,000	118	24.6%
	Over 40,000	25	5.2%
City type	First-tier and new first-tier	301	62.7%
	Second-tier	45	9.4%
	Third-tier and below	134	27.9%

**Table 4 foods-13-02239-t004:** Correlation analysis and reliability and validity test.

Variant	AVE	CR	SN	ATT	PBC	SI	CI	IN	BE
SN	0.647	0.785	0.804						
ATT	0.667	0.889	0.556 **	0.817					
PBC	0.652	0.849	0.424 **	0.374 **	0.808				
SI	0.591	0.896	0.482 **	0.501 **	0.459 **	0.769			
CI	0.540	0.824	0.406 **	0.462 **	0.351 **	0.361 **	0.735		
IN	0.712	0.881	0.583 **	0.556 **	0.498 **	0.615 **	0.556 **	0.844	
BE	0.836	0.910	0.593 **	0.529 **	0.505 **	0.602 **	0.529 **	0.714 **	0.914

Note: ** *p* < 0.01.

**Table 5 foods-13-02239-t005:** Model’s goodness-of-fit.

Indices	Metric	Threshold	Result
CMIN/DF	2.217	1 < NC < 3	Accept
GFI	0.918	>0.80	Accept
AGFI	0.894	>0.80	Accept
CFI	0.959	>0.90	Accept
IFI	0.959	>0.90	Accept
RMR	0.034	<0.05	Accept
RMSEA	0.05	<0.08	Accept

**Table 6 foods-13-02239-t006:** Results of hypothesis testing.

Hypothesis	Path	Standardized Path Coefficient (β)	T-Value	Result
H1a	SN→IN	0.163 **	3.030	Supported
H2a	ATT→IN	0.257 ***	4.973	Supported
H3a	PBC→IN	0.131 **	2.866	Supported
H4a	SI→IN	0.264 ***	5.375	Supported
H5a	CI→IN	0.230 ***	4.933	Supported
H1b	SN→BE	0.152 **	3.019	Supported
H2b	ATT→BE	0.154 **	3.123	Supported
H3b	PBC→BE	0.111 **	2.612	Supported
H4b	SI→BE	0.173 ***	3.686	Supported
H5b	CI→BE	0.132 **	2.953	Supported
H6	IN→BE	0.293 ***	4.844	Supported

Note: ** *p* < 0.01 and *** *p* < 0.001.

**Table 7 foods-13-02239-t007:** Fitness of multiple sets of models.

Indices	Gender	Age	Education Level	Monthly Household Income	City Type
CMIN/DF	1.696	1.682	1.718	1.774	1.749
GFI	0.882	0.884	0.881	0.880	0.895
AGFI	0.847	0.849	0.845	0.844	0.864
CFI	0.953	0.955	0.952	0.947	0.955
IFI	0.954	0.955	0.953	0.948	0.956
RMR	0.041	0.040	0.042	0.042	0.038
RMSEA	0.038	0.038	0.039	0.040	0.037

**Table 8 foods-13-02239-t008:** Results of multi-group analyses.

Path	Gender		Age		Education Level	
	Male	Female	Young	Middle-Aged and Elderly	Low	High
SN→IN	0.113	0.266 **	0.119	0.211 ***	0.266 ***	−0.009
ATT→IN	0.228 ***	0.256 **	0.189 *	0.305 ***	0.260 ***	0.268 ***
PBC→IN	0.142 *	0.133	0.131	0.128 *	0.111 *	0.170 *
SI→IN	0.333 ***	0.170 *	0.191 *	0.322 ***	0.327 ***	0.208 *
CI→IN	0.244 ***	0.215 **	0.308 ***	0.170 **	0.225 ***	0.261 ***
SN→BE	0.132 *	0.147	0.234 ***	0.075	0.082	0.267 **
ATT→BE	0.142 *	0.162 *	0.190 **	0.090	0.186 **	0.121
PBC→BE	0.097	0.105	0.128 *	0.099	0.137 *	0.082
SI→BE	0.221 *	0.154 *	0.159 *	0.163 *	0.202 **	0.118
CI→BE	0.123	0.124 *	0.112	0.142 *	0.161 **	0.079
**Path**	**Monthly Household Income**			**City** **Classification**		
	**Low**	**High**		**First-Line**	**Non-First-Tier**	
SN→IN	0.220 **	0.114		0.042	0.215 **	
ATT→IN	0.313 ***	0.248 **		0.280 ***	0.249 ***	
PBC→IN	0.150 *	0.105		0.120	0.111	
SI→IN	0.321 ***	0.206 **		0.343 ***	0.293 ***	
CI→IN	0.224 ***	0.239 ***		0.238 ***	0.134 *	
SN→BE	0.169	0.130 *		0.068	0.227 ***	
ATT→BE	0.093	0.149 *		0.167 **	0.127 *	
PBC→BE	0.031	0.149 **		0.139 *	0.029	
SI→BE	0.079	0.212 ***		0.156 **	0.234 ***	
CI→BE	0.096	0.132 *		0.152 *	0.090	

Note: * *p* < 0.05, ** *p* < 0.01, and *** *p* < 0.001.

**Table 9 foods-13-02239-t009:** Analysis of the moderating effect of price fairness.

Variant	Purchase Frequency			Purchase Proportion		
	Model 1	Model 2	Model 3	Model 1	Model 2	Model 3
IN	0.908 **	0.831 **	0.851 **	0.882 **	0.778 **	0.816 **
PF		0.171 **	0.174 **		0.230 **	0.235 **
IN × PF			0.119 **			0.224 **
Coefficient of determination	0.651	0.67	0.677	0.508	0.536	0.558
F-value	760.622	413.134	284.239	421.139	235.469	171.163

Note: ** *p* < 0.01.

**Table 10 foods-13-02239-t010:** Analysis of the moderating effect of purchase convenience.

Variant	Purchase Frequency			Purchase Proportion		
	Model 1	Model 2	Model 3	Model 1	Model 2	Model 3
IN	0.908 **	0.836 **	0.845 **	0.882 **	0.801 **	0.829 **
PC		0.166 **	0.176 **		0.185 **	0.219 **
IN × PC			0.072			0.229 **
Coefficient of determination	0.651	0.668	0.671	0.508	0.526	0.551
F-value	760.622	409.949	276.284	421.139	225.707	165.815

Note: ** *p* < 0.01.

**Table 11 foods-13-02239-t011:** Multi-category regression results.

Variant	Low Consistency (LC)	High Consistency (HC)
SN	1.054 ** (3.469)	1.610 ** (4.47)
ATT	1.572 ** (4.846)	2.830 ** (6.758)
PBC	0.704 * (2.335)	1.367 ** (3.936)
SI	0.894 ** (3.608)	2.136 ** (6.151)
CI	0.546 * (2.475)	1.322 ** (3.951)
PF	−0.24 (−0.876)	0.15 (0.457)
PC	0.42 (1.531)	0.5 (1.513)

Note: * *p* < 0.05, ** *p* < 0.01.

**Table 12 foods-13-02239-t012:** Multinomial grouping regression results.

Variant	Gender				Age				Educational Level			
	Male		Female		Youth		Middle and Old Age		Low		High	
	LC	HC	LC	HC	LC	HC	LC	HC	LC	HC	LC	HC
SN	0.737 (1.918)	1.489 ** (3.002)	1.336 ** (2.852)	1.899 ** (3.584)	1.555 ** (2.631)	2.260 ** (3.553)	0.922 * (2.282)	1.483 ** (2.721)	1.032 ** (3.013)	1.230 ** (2.688)	2.114 * (2.052)	3.078 ** (2.899)
ATT	1.525 ** (3.601)	3.205 ** (5.327)	1.576 ** (3.073)	2.683 ** (4.385)	2.189 ** (3.473)	3.273 ** (4.631)	1.207 ** (3.094)	3.179 ** (4.905)	1.097 ** (3.283)	2.858 ** (5.229)	5.017 * (2.504)	6.108 ** (3.002)
PBC	1.205 ** (3.075)	1.913 ** (4.201)	0.710 (1.677)	1.572 ** (3.214)	1.630 ** (3.188)	2.392 ** (4.256)	0.621 (1.805)	1.481 ** (3.411)	0.823* (2.525)	1.836 ** (4.024)	2.489 * (2.268)	3.220 ** (2.880)
SI	0.537 (1.484)	1.967 ** (3.997)	1.118 ** (3.019)	2.351 ** (4.609)	1.493 ** (3.407)	2.886 ** (5.338)	0.453 (1.243)	2.100 ** (3.571)	0.741 ** (2.719)	3.243 ** (5.549)	3.443 * (2.205)	4.201 ** (2.650)
CI	0.331 (1.084)	1.072 * (2.372)	0.831 * (2.351)	1.734 ** (3.341)	1.052 ** (2.697)	1.877 ** (3.637)	0.282 (1.014)	1.197 * (2.416)	0.258 (1.001)	1.286 ** (2.646)	3.433 ** (2.700)	3.129 ** (3.259)
**Variant**	**Monthly Household Income**						**City Type**					
	**Low**			**High**			**First-tier**			**Second-Tier and Below**		
	**LC**		**HC**	**LC**		**HC**	**LC**		**HC**	**LC**		**HC**
SN	0.897 * (2.155)		1.126 * (2.190)	1.490 ** (2.898)		2.327 ** (4.105)	0.429 (0.976)		0.985 (1.915)	2.300 ** (3.053)		2.988 ** (3.688)
ATT	1.722 ** (3.477)		3.025 ** (4.438)	1.335 ** (2.905)		2.806 ** (4.950)	1.622 ** (3.849)		2.895 ** (5.389)	2.134 ** (3.122)		3.607 ** (4.449)
PBC	0.527 (1.566)		1.320 ** (2.933)	1.570 ** (3.065)		2.345 ** (4.247)	0.255 (0.999)		1.012 * (2.432)	2.277 ** (3.199)		3.115 ** (4.077)
SI	0.965 ** (2.949)		2.594 ** (4.518)	0.678 (1.511)		1.943 ** (3.645)	1.028 ** (2.884)		2.057 ** (5.241)	1.272 ** (2.597)		2.447 ** (3.932)
CI	0.185 (0.661)		1.320 * (2.512)	1.198 ** (2.888)		1.881 ** (3.620)	0.372 (1.135)		1.393 ** (2.951)	1.340 ** (2.699)		1.943 ** (3.100)

Note: * *p* < 0.05 and ** *p* < 0.01.

**Table 13 foods-13-02239-t013:** Assessment indicators for each model.

Model	Training Set				Testing Set			
	Accuracy	Recall	Precision	F1	Accuracy	Recall	Precision	F1
Decision tree	0.955	0.955	0.955	0.955	0.748	0.748	0.747	0.747
Random forest	0.983	0.983	0.983	0.983	0.821	0.821	0.82	0.818
AdaBoost	0.777	0.777	0.779	0.774	0.797	0.797	0.794	0.792
Logistic regression	0.812	0.812	0.813	0.812	0.805	0.805	0.804	0.804

## Data Availability

The original contributions presented in the study are included in the article, further inquiries can be directed to the corresponding author.

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
