# Peer review of "Bridging the Intention–Behavior Gap in Organic Food Consumption: Empirical Evidence from China"

_foods, 2024, doi:10.3390/foods13142239_

Round 1
Reviewer 1 Report
Comments and Suggestions for Authors
Dear authors,
The manuscript entitled Bridging the intention-Behavior Gap in Organic Foods Consumption: Empirical evidence from China is relevant to the journal's aims and scope. The manuscript is very interesting, but it still has some issues that need to be addressed:
The abstract lacks specific details about the study's methodology, sample size, and data sources, which are essential for understanding the scope and reliability of the research.
The introduction is overly long and contains redundant information, making it difficult for readers to grasp the main points quickly. The same ideas, such as the intention-behavior gap and the importance of organic food consumption, are repeated multiple times. Additionally, the introduction attempts to cover too many aspects, including environmental statistics, government policies, theoretical frameworks, and empirical findings, without sufficiently focusing on the specific problem the study addresses.
While the introduction provides some background, it fails to clearly set the stage for the specific research problem. More context on why the intention-behavior gap is particularly significant in the realm of organic food consumption would be helpful.
The literature review does not provide sufficient detail about the theories and studies it references. For instance, the explanations of the theory of planned behavior, social interactionism, and social cognitive theory are too brief and lack depth. Additionally, the literature review repeats certain points without adding new insights, such as the impact of social interaction on consumption behavior, which is mentioned multiple times without further elaboration.
Furthermore, while the hypotheses are listed, their integration into the overall discussion is unclear. The relationship between the theoretical framework and the hypotheses should be more explicitly articulated. The review also fails to critically analyze the existing literature, merely summarizing studies without discussing their strengths, weaknesses, or how they contribute to the current research gap.
The recommendations are vague and lack practical steps for implementation. For instance, suggestions for marketers to "implement diverse promotional activities" or for consumers to "separate organic food from other food products" lack specific strategies.
While limitations regarding data volume, consistency levels, and sample size are mentioned, there is no discussion on their impact or how future research can address these issues.
The section on machine learning and its role in prediction is poorly integrated into the conclusions and recommendations. The practical implications should be more explicitly discussed and connected to the recommendations.
The conclusions make broad generalizations about different consumer groups without detailed evidence from the study, oversimplifying the diversity within these groups.
The conclusions do not critically analyze the results or acknowledge potential limitations and biases, such as cultural differences, regional variations, or socioeconomic factors.
Reviewer 2 Report
Comments and Suggestions for Authors
The paper deals with a very interesting topic and is really well structured and clear.
Some points needed to improve the article:
1) better explain the reasons why you arrived at the formulation of that theoretical model, in light of the background produced on the different theories proposed. Add references on this.
2) the paper addresses an important gap, the literature should be taken into consideration, to support the interpretation of the results, which tells us about a citizen-consumer gap, which refers to an attitude-behavior gap. This gap is present for many research areas, in particular organic agriculture.
3) the references are really scarce, please integrate some literature to support the results
4) explain in the conclusions how these results could offer insights to the policy network in China
Comments on the Quality of English Languageminor
Round 2
Reviewer 1 Report
Comments and Suggestions for Authors
Dear authors,
Thank you for accepting suggestions for improving your manuscript.
The manuscript meets all requirements for publication in the scientific journal Foods.
Author Response
Thank you very much for your acceptance of my manuscript.